# Theoretical Prediction of Mechanical Strength and Desalination Performance of One-Atom-Thick Hydrocarbon Polymer in Pressure-Driven Separation

**DOI:** 10.3390/polym11081358

**Published:** 2019-08-16

**Authors:** Shuangqing Sun, Fei Shan, Qiang Lyu, Chunling Li, Songqing Hu

**Affiliations:** 1School of Materials Science and Engineering, China University of Petroleum (East China), Qingdao 266580, China; 2Institute of Advanced Materials, China University of Petroleum (East China), Qingdao 266580, China

**Keywords:** mechanical strength, membrane-based separation, two-dimensional hydrocarbon polymer, water desalination, molecular simulation

## Abstract

One-atom-thick materials hold promise for the future of membrane-based gas purification and water filtration applications. However, there are a few investigations on the mechanical properties of these materials under pressure-driven condition. Here, by employing molecular simulation techniques and continuum mechanics simulation, we investigate the mechanical strength of two-dimensional hydrocarbon polymers containing sub-nanometer pores with various topologies. We demonstrate that the mechanical strengths of the membranes are correlated with their pore sizes and geometries. In addition, when the pore size of substrates is controlled within a reasonable range, all of the membrane candidates can withstand the practical hydraulic pressure of few megapascal. The studied materials also exhibit better seawater desalination performance as compared to the traditional polymeric reverse osmosis membrane. This work presents a new route to design new separation membrane, and also propose a simulation method to evaluate the mechanical strength and desalination performance.

## 1. Introduction

Membrane-base separation technology has been applied for the purification of biofuel extraction, water desalination, and sewage treatment owing to their characteristics such as high efficiency, energy conservation, and environmental friendliness [1,2,3,4,5,6]. To date, graphene oxides [7], covalent organic frameworks (COFs) [8,9], metal organic frameworks (MOFs) [10,11,12], zeolite [13], carbon nano tube (CNT) [14,15], and other advanced materials are considered as separation membrane materials. Among numerous membrane materials, nanoporous ultrathin-film materials, such as one-atom-thick porous graphene (PG), have drawn considerable attention in the scientific community for their potential as the next generation of separation membrane materials [16,17,18,19,20,21,22,23]. To date, PG has been prepared by drilling nanopores in original graphene sheets, denoted as the so-called top-down approach [24,25,26,27].

Recently, a great deal of two-dimensional (2D) hydrocarbon polymers with intrinsic regularly aligned pore array and precisely controlled pore size, such as graphdiyne, triphenylene graphdiyne (TP-GDY) [28], 2D-polyphenylene (2D-PP) [29], and 2D-conjugated aromatic polymer (2D-CAP) [30], have been experimentally achieved using the commonly known bottom-up method. Bieri et al. successfully synthesized 2D-PP network (also denoted as PG because of the structural analogy with hydrogenated PG) by cross-coupling macrocycle cyclohexa-m phenylene on a silver surface [26]. This synthesis method can effectively adjust the pore size. For instance, a series of analogues structures has been precisely designed by replacing biphenyl-like subunits of 2D-PP with para-terphenyl (TP) like bonding patterns along the hexagonal directions. Depending on the number of TP-like units (*X*, *X* = 0~3), the resulting structures (i.e., denoted as PG-TP*X* hereafter) possess pores with different sizes of varying number of periodically missing hexagons. Because of the ideal atomic-unit thickness of such materials, together with a precise control of the pore size, these materials are potentially used in liquid-phase and/or gas-phase separation applications. As an example, Schrier et al. have computationally demonstrated that PG-TP1 has the ability as the high-performance paraffin/olefin separation membrane [31].

From a practical viewpoint, the separation membrane must be pressurized by gases and liquids. In the separation of methane and carbon dioxide, separation pressure can be up to 50 bar [2]. In the field of sewage treatment, 60 bar pressure is usually required to achieve excellent treatment results [3]. The typical pressure of membrane-based materials for desalination of seawater is 55 bar. This means they must bear certain pressure when membrane materials are used for separation applications. In other words, membrane materials require enough mechanical strength. However, there are few investigations on whether ultrathin-film materials are strong enough to maintain its mechanical integrity under such high hydraulic pressure inherent to the separation process as mentioned above. In addition to the favorable mechanical properties of membrane materials, a substrate layer is always used to support the active layer (i.e., the separation membrane materials) in the practical pressure-driven separation system, bearing much permeating load while distributing the pressure from permeating media onto the active layer materials [32,33,34]. Therefore, the mechanical strength of PG-TP*X* membranes and their supporting substrates should also be taken into consideration although the literature on this topic is scarce.

In this study, by employing the state-of-the-art molecular simulation techniques, we investigated the mechanical stability of one-atom-thick PG-TP*X* membranes in the specific context of pressure-driven separation process. In addition, to our knowledge, there are few studies on reverse osmosis (RO) desalination of these membranes, so we also explored their potential as RO membranes for water desalination application. In this paper, the mechanical strength of the PG-TP*X* membrane was investigated, and the conditions and current application status of PG-TP*X* membrane were discussed. This research will be helpful for the designing of new RO membranes and further the understanding of their mechanical properties at the atomic level.

## 2. Computational Details

### 2.1. Tensile Tests

Classical molecular dynamics (MD) simulations were employed to perform biaxial (for the determination of fracture stress, elongation at break, and Young’s modulus) and uniaxial (for the determination of Poisson’s ratio) tensile tests of PG-TP*X* membranes. The number of TP-like unit (defined as *X*) ranges from 0 to 3, leading to membrane structures with different pore sizes and periodically missing hexagons (Figure 1a–d). The membrane models with an initial dimension of approximately 100 × 100 Å^2^ in the *x*–*y* plane were located in the middle of the simulation box, while a vacuum slab was added along the *z* direction to make a box with the height of 80 Å. Periodic boundary conditions were applied in all three directions to eliminate the boundary effect of the atomic small-size membrane. Figure 1e,f shows the tensile test along armchair and zigzag directions of the membrane, respectively. The *z* direction remains unchanged during tension. All the above operations were carried out by Materials Studio.

To properly describe the bond association/disassociation events during the tensile process, reactive force field (ReaxFF) [35] was adopted. Moreover, to make sure our simulations are not biased by different potential versions, two different sets of parameters (i.e., Budzien et al. and Strachan et al.) [36,37] were accepted. The mechanical parameters (i.e., tensile stress, fracture strain, Young’s module, and Poisson’s ratio) from the two potentials were in excellent agreement (see Appendix A in Appendix A for the details). Before loading the process of strain, structural optimizations of the initial PG-TP*X* materials were performed to release the in-plane residual stresses. Biaxial strain tests were performed in the NVT ensemble using a Nosé–Hoover thermostat [38] at 300 K with a damping factor of 100 time steps, whereas a Nosé–Hoover barostat at zero pressure was additionally applied on the non-enforced-strained boundary in the uniaxial strain tests (i.e., in the NPT ensemble) with a damping factor of 1000 time steps. A strain increment per 5000 steps at a rate of 5 × 10^−7^ fs^−1^ was applied for both types of tensile simulations. The time step was set to 0.1 fs. The stress at a given strain was obtained by averaging the stress in the last 2500 steps before the next strain increment. All continuum mechanics simulations were carried out using the open-source large-scale atomic/molecular massively parallel simulator (LAMMPS)package [39].

### 2.2. Reverse Osmosis Desalination 

We constructed a combined model to assess the desalination performance of the membrane, i.e., water permeability and salt rejection. Figure 2 exhibits the model which consists of two rigid graphene pistons on the two ends, a membrane in the middle, and salt water and fresh water between the membrane and the two pistons. The membrane and the two pistons have been relaxed by Dmol3 using density functional theory (DFT) (lattice parameters and parameter setups during DFT simulations can be found in Appendix A). The box dimension in the *x*–*y* plane is 45 × 35 Å^2^, and a large vacuum slab has been added to the *z* direction to provide enough mobile space for the piston wall. Periodic boundary conditions were applied in all three directions. 18 Na^+^/Cl^−^ ion pairs were randomly distributed into 1800 water molecules to construct the salt water region, while the fresh water region was packed with 800 water molecules. Seawater desalination models were established in Material Sudio.

The nonbonding interactions between pairwise atoms were described by 6–12 Lennard-Jones (L-J) potentials and Coulombic interactions based on point charge models. The L-J potential parameters and charges for the systems are summarized in Appendix A. The parameters for L-J potentials were taken from the DREIDING force field [40], and the pairwise interactions were truncated by a cutoff radius 12 Å. The atomic partial charges for Coulombic interactions were assigned by the charge equilibration (QEq) method [41], and the long-range interactions were optimized by the particle–particle particle–mesh (PPPM) algorithm with a precision of 10^−6^ e. Water molecules were modeled by the rigid SPC/E mode l [42] with the SHAKE [43] algorithm. The nonbonding potentials for salt ions and piston atoms were developed by Joung et al. [44] and Werder et al. [45], respectively.

The system was first relaxed by an energy minimization process. Then, canonical ensemble (i.e., NVT) MD simulations at 300 K were performed for the equilibrium and production processes. The equilibrium process took 0.5 ns, during which a hydrostatic pressure of 1 atm was applied on both pistons (*P_f_* and *P_p_*) toward the middle membrane, to bring the liquid density closer to 1.0 g cm^−3^. For the production process, a transmembrane pressure *P_f_* from 5 MPa to 70 MPa was applied on the feed piston to push the salt water across the membrane, during which the middle membrane was kept constrained. The Nosé–Hoover thermostat [38] with 100 time steps damping constants (i.e., 100 fs) was used to control system tmperature. The time step was set to 1 fs. Additional computation details can be found in SI. We adopted five independent above-mentioned simulations to get the statistic reasonable results. All simulations in this work were carried out using the open-source LAMMPS package [39].

## 3. Results and Discussion

### 3.1. Mechanical Strength of PG-TPX Membranes

#### 3.1.1. Mechanical Parameters of PG-TPX Membranes

Stress–strain curves from biaxial tensile tests of PG-TP*X* membranes by MD simulations are shown in Figure 3. Young’s modulus (*E_M_*), fracture stress (*σ_M_*), and elongation at break (*ε_M_*) obtained from biaxial tensile tests are shown in Table 1. First, PG-TP*X* membranes exhibit elastic behavior prior to fracture, although the relation between stress and strain is nonlinear. This observation is consistent with the previously reported work on the mechanical properties of grapheme [39]. Second, PG-TP*X* membranes exhibit different fracture stress, and break at different strains (i.e., elongation at break, *ε_M_*). Taking PG-TP2 as a representative instance, the stress gradually increases with the increase of strain, and the membrane finally fractures when the strain reaches to 11.58% and the corresponding stress reaches to 43.012 MPa. Figure 4 shows the stress distribution in PG-TP2 membrane with increasing biaxial stress. It is seen that the entire membrane is in a state of plane stress prior to fracture. The first bond to break is the C–C single bond that connects two adjacent benzene rings. This indicates that failure in PG-TP2 is characterized by brittle fracture at the site of the C–C single bond.

However, because of the different pore sizes and structures of the membrane, the structural changes of different membranes in the tensile process are different, resulting in different mechanical parameters of the membrane. Table 1 shows the Young’s modulus (*E_M_*) and elongation at break (*ε_M_*) for four membranes. It is seen that *ε_M_* increases as the number of TP-like units (i.e., pore area) increases. However, *E_M_* of PG-TPX sheet does not increase with increasing pore area. We suppose that this phenomenon is attributed to the different pore structures. The pore of PG-TP1 is biased toward a rectangle structure, while PG-TP2 is biased toward a diamond structure. When tension is applied to the membrane, the force direction of the C–C single bond connecting the benzene ring is different. In the armchair’s force direction, the C–C single bond on PG-TP1 is tensioned horizontally, while that the bond of PG-TP2 is at an angle to the armchair direction. So we predicted this makes PG-TP2 hard to be deformed and thus exhibits stronger mechanical strength.

In the following investigation on the substrate pore size and the mechanical properties of the practical pressure-driven separation membrane system, the Poisson’s ratio of the membrane material is required, so we also conducted uniaxial tensile simulation for different membrane materials. Since the membrane has different structures along the armchair and zigzag directions, the Poisson’s ratio depends on the tension direction. If the Poisson’s ratio in the tension direction is smaller, the strain in this direction will be larger than that in the unstretched direction, and the membrane is more likely to fracture. Therefore, we choose the smaller Poisson’s ratio from the two tensile directions as the membrane’s Poisson’s ratio, which reflects the critical fracture strength of the membrane. The Poisson’s ratios of membrane materials obtained from uniaxial tensile tests are listed in Table 1.

#### 3.1.2. Optimization of Substrate Pore Size

It has been stated that in the practical pressure-driven separation system a substrate layer is always used to support the active layer (i.e., the separation membrane materials). In Section 3.1.1, we have acquired the Young’s modulus (*E_M_*), fracture stress (*σ_M_*), and elongation at break (*ε_M_*) from biaxial tensile tests and Poisson’s ratio (*ν*) from uniaxial tensile tests (Table 1). Through the above parameters, we continued to analyze the actual stress of membrane materials when they are subjected to hydraulic pressure. The relationship between the maximum hydraulic pressure that the separation membrane can bear and the pore diameter of the substrate can be described by a microscopic equation [46]. Assuming that the homogeneous separation membrane is clamped at the substrate pore edges and no residual stress exists prior to tensile loading, the actual stress of membrane, *ρ* is expressed as:
(1)$$\rho ={\left(\frac{2}{3}\frac{E_{M}}{1.026-0.793v_{M}-0.233{v_{M}}^{2}}\right)}^{1/3}{\left(\frac{\Delta PR}{4d_{M}}\right)}^{2/3}$$
where *E_M_* and *v_M_* are the Young’s modulus and Poisson’s ratio of the membrane, respectively, *R* is the substrate pore radius, *d_M_* is the thickness of the membrane, and Δ*P* is the applied hydraulic pressure [46]. Considering the actual stress shown in above equation reaches to the fracture stress of each membrane material, we can obtain the relationship between the maximum hydraulic pressure that the membrane material can withstand (Δ*P*) and the pore size of the substrate (*R*) (Figure 5a). It can be seen that Δ*P* decreases with the increase of *R*. When the PG-TPX membrane materials were attached to the substrate with the same pore size, PG membrane can withstand the maximum hydraulic pressure, followed by PG-TP2, PG-TP1, and PG-TP3. From Figure 5a, we can also obtain the conclusion that when the same hydraulic pressure (5 Mpa) is applied to different PG-TPX separation membrane systems, the allowable pore size of substrate for PG, PG-TP1, PG-TP2, and PG-TP3 is 2.3 μm, 2.8 μm, 3.8 μm, and 5.3 μm, respectively. 

In addition, PG-TP2 membrane with the fracture stress of 43.012 GPa was taken as an example to probe the relation between its actual stress and substrate pore radius under different external hydraulic pressures (Figure 5b). Generally, it can be seen that the allowable internal stress of the separation membrane (i.e., fracture stress, *σ_M_*, in this work) increases with the increase of the substrate pore size. In addition, for PG-TP2 membrane with the fracture stress of 43.012 GPa, the allowable substrate pore decreases with the increase of hydraulic pressure. When the hydraulic pressure (Δ*P*) reaches 10 MPa, the allowable substrate pore size is approximately 1.8 μm. However, if we increase the pore size of the substrate to 3.8 μm, the allowable hydraulic pressure can be greatly decreased to 5 MPa (similar to the pressure value used in the practical separation process). The other three membrane materials have the same trend, as shown in Appendix A. Therefore, as long as the substrate pore size is controlled within a reasonable range, all four materials will not be broken by the typical hydraulic pressure in the process of desalination. In other words, the theoretical predictions indicate that 2D hydrocarbon polymers PG-TP2 designed in this work supported by the substrate materials with the appropriate pore size can withstand the hydraulic pressure for practical separation application without brittle fracture.

Nowadays, polysulfone substrate has been extensively used as a supporting substrate material for water desalination and gas purification. Nakao et al. [47] analyzed the substrate pore sizes of polysulfone and found that the average pore radius of most polysulfone substrates is ~0.2 μm, which is much smaller than the calculated allowable pore size (1.8 μm) corresponding to the fracture stress (i.e., 43.012 GPa for PG-TP2 membrane). Therefore, it can be proposed that polysulfone substrate with common pore size is suitable as the supporting substrate for the separation membranes studied in this work in terms of mechanical stability.

In this part, we investigated the mechanical strength of two-dimensional hydrocarbon polymers containing sub-nanometer pores with various topologies. We demonstrated that the mechanical strength of the membranes is correlated with their pore sizes and geometries. In addition, when the pore size of the substrate is controlled within a reasonable range, all the studied PG membranes can withstand the practical hydraulic pressure of few megapascals. Results in relation to above discussions have not been previously reported.

### 3.2. Desalination Performance of PG-TPX Membranes

It has been shown that the mechanical strength of PG-TP*X* materials can meet the requirements of pressure-driven separation application. It remains, however, crucial to further investigate the separation performance of PG-TP*X* membranes. Specifically, we consider their applications in water desalination in this study. To understand the desalination performance of PG-TP*X* membranes, MD simulations were adopted to predict water permeability and salt rejection rate (see Computational details for details).

In this work, water molecules on both sides of the membrane are consistently exchangeable under an external hydraulic pressure, and it results in a net water flow from the feed side to the permeate side. The water fluxes for each membrane pore as a function of applied external hydraulic pressure are summarized in Figure 6a. Theoretically, a water molecule has a kinetic diameter of approximately 2.8 Å [48], and therefore less likely to permeate through the membrane with a pore diameter smaller than or close to this value. Therefore, as expected, water molecules are unable to cross the PG and PG-TP1 membranes that possess relatively smaller pores (pore diameters of 0.948 Å and 1.558 Å, respectively). Even at a theoretically high pressure of 70 MPa, the water flux of these two membranes is essentially zero. By contrast, water molecules can filter through both PG-TP2 and PG-TP3 membranes with pore diameter of 4.555 Å and 8.425 Å, respectively, leading to a nonzero water flux.

It has been reported by many studies that the water flux is linear-related to the applied hydraulic pressure, and the typical pressure of RO units (usually only a few MPa) can be extrapolated from this flux-pressure curve [49]. As we all know, osmotic pressure is the minimum pressure applied on the high concentration side in a semipermeable membrane having different aqueous solution concentrations on both sides to prevent water penetrating from the low concentration side to the high concentration side. In this work, the flux-pressure curve shown in Figure 6a crosses the *x*-axis at an applied pressure of 2.69 MPa, which is thought to be the osmotic pressure of the feed side in our simulated separation membrane system. Precisely, this result is consistent with the theoretical permeation values (i.e., 2.77 MPa) calculated from the solution concentration. This proves the accuracy of the calculation method implemented in this work. Moreover, water permeability is defined as the slope of the aforementioned flux–pressure curves per membrane area per applied pressure (i.e., L·cm^−2^·day^−1^·MPa^−1^), as shown in Figure 6b inset. PG-TP2 and PG-TP3 membranes appear to possess excellent water permeability of 45.12 and 287.66 L·cm^−2^·day^−1^·MPa^−1^, respectively. Compared with commercial seawater RO, brackish RO, and high-flux RO membranes, the PG-TP2 and PG-TP3 are theoretically predicted to possess four and five orders of magnitude higher permeability [50].

To evaluate the desalination performance of PG-TP*X* membrane candidates, in addition to water permeability it is essential to analyze the salt rejection. The calculated salt rejection for each membrane as a function of the hydraulic pressure is presented in Figure 6b. The salt rejection of PG-TP2 membrane is close to 100% (except the salt rejection of 99% at a theoretically high pressure of 70 MPa), which can be explained by the size of membrane pores and hydrated salt ions. The sizes of hydrated Na^+^ (7.16 Å) and Cl^−^ (6.64 Å) are larger than the effective pore size (4.56 Å for PG-TP2), thus it is a virtual impossibility for the hydrated ions to pass through this membrane, leading to an approximately full salt rejection. For PG-TP3 membrane with the effective pore size of 8.425 Å, its salt rejection is about 80–90% at the pressure range of 10–25 MPa.

Overall, compared with the polymeric RO membrane with the water permeability of ~0.17 L·cm^−2^·day^−1^·MPa^−1^, PG-TP2 evidently achieves two orders of magnitude higher permeability and comparable salt rejection to those promising ultrathin-film membranes that have been computationally identified [21,22]. Therefore, the simulation results indicate that it can potentially act as a high-permeability RO membrane for one-step water desalination. Moreover, owing to the relatively lower salt rejection, PG-TP3 can provide opportunities in brackish water desalination or multi-stage seawater desalination.

## 4. Conclusions

In this work, we used molecular dynamics technique combined with a continuum mechanical model to study the mechanical strength of PG-TP*X* materials. It is found that, the mechanical strength of the membranes is related to their pore sizes and topologies. As the pore size increases, the Young’s modulus and fracture stress of the membrane materials decrease gradually, but PG-TP2 has better Young’s modulus and fracture stress. When the substrate with the same pore size is used, PG can withstand the biggest hydraulic pressure, followed by PG-TP2. As long as the pore size of supporting substrate is controlled within a reasonable range, all studied materials can withstand the hydraulic pressure under practical separation conditions. Moreover, the simulation results predict that one-atom-thick hydrocarbon polymers have better water desalination performance compared with the traditional polymeric reverse osmosis membranes. For instance, PG-TP2 is predicted to offer an exceptional water permeability of approximately 60 L m^−2^ h^−1^ bar^−1^ and a salt rejection rate of 100%. Overall, this study is instrumental in designing new membranes and evaluating their properties for pressure-driven separation applications.

## Figures and Tables

**Figure 1 polymers-11-01358-f001:**
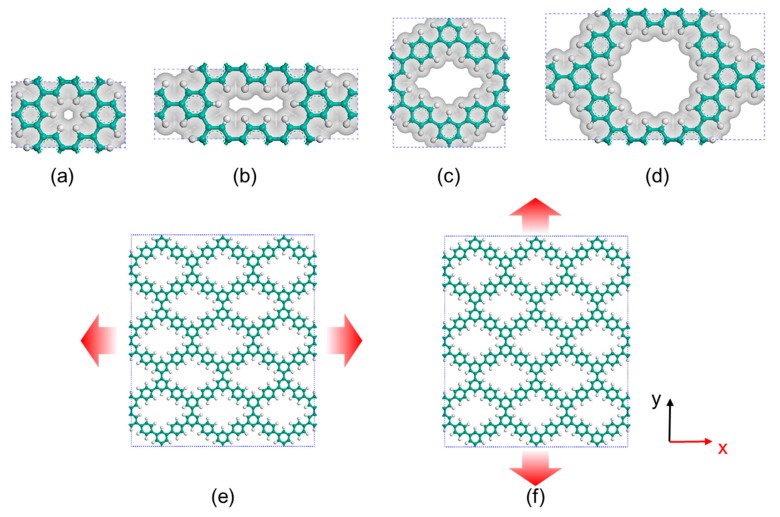
(**a**–**d**) Atomic structures and pore topology of PG-TP*X* (*X* = 0, 1, 2, and 3, with pore areas of 0.8171, 6.6796, 28.4535, and 63.9226 Å^2^, respectively). Tensile model of a representative structure (i.e., PG-TP2) along the armchair (**e**) and zigzag (**f**) directions. Atom color: carbon-cyan, hydrogen-white. Van der Waals surfaces are shown in transparent cyan.

**Figure 2 polymers-11-01358-f002:**
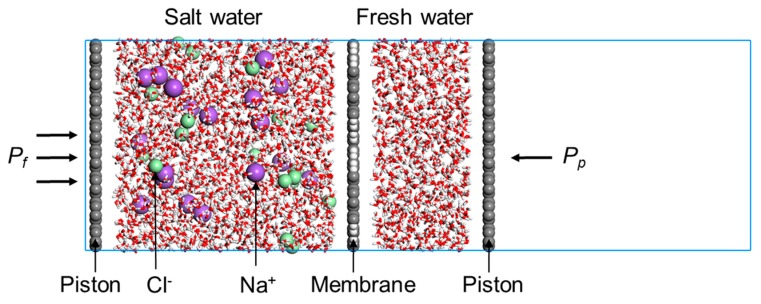
Schematic view of the reverse osmosis (RO) desalination system. PG-TP2 is used as the active membrane layer. Atom color: carbon-gray, hydrogen-white, oxygen-red, sodium-purple, and chloride-green.

**Figure 3 polymers-11-01358-f003:**
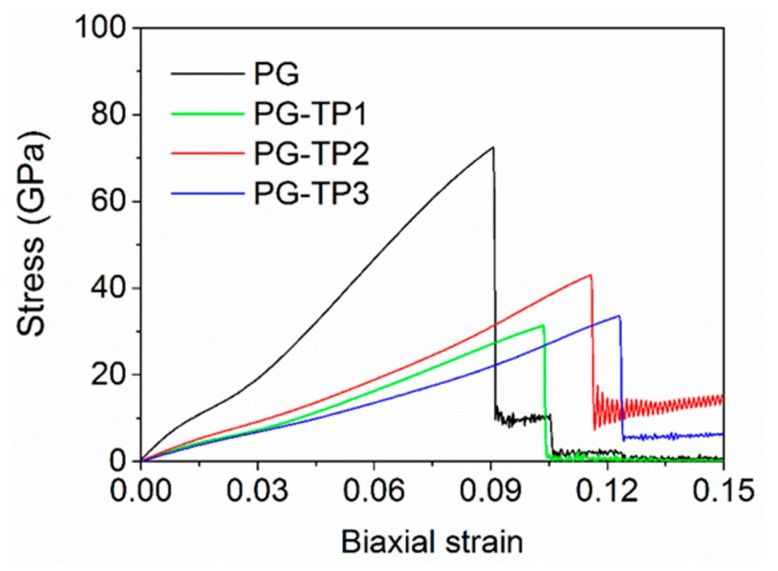
Stress–strain curves obtained from biaxial tensile tests.

**Figure 4 polymers-11-01358-f004:**
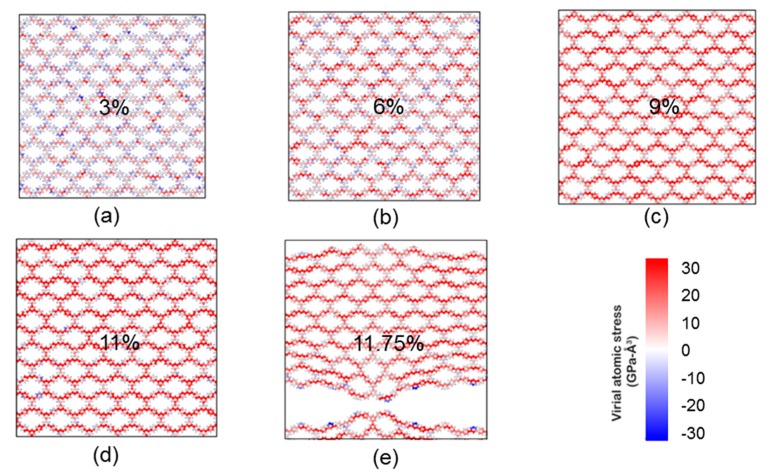
Stress distribution in PG-TP2 sheet with increasing biaxial stress. The stress at each atom is represented by its color with red regions corresponding to the areas of highest stress (see the color bar). The strain in the figure is (**a**) 3%, (**b**) 6%, (**c**) 9%, (**d**) 11%, and (**e**) 11.75%.

**Figure 5 polymers-11-01358-f005:**
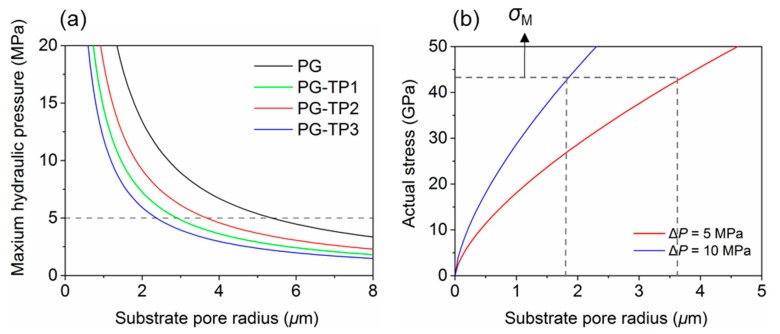
(**a**) Maximum hydraulic pressure as a function of substrate pore radius. (**b**) Stress in membrane as a function of substrate pore radius under different hydraulic pressure (PG-TP2).

**Figure 6 polymers-11-01358-f006:**
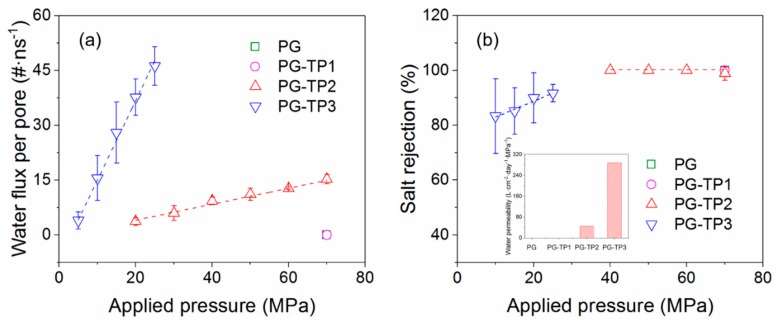
Water flux and salt rejection of porous graphene (PG) membranes. (**a**) Water flux and (**b**) salt rejection as a function of applied pressure, inset shows water permeability of each PG membrane. Lines serve as guides to the eyes.

**Table 1 polymers-11-01358-t001:** Mechanical parameters of PG-TPX sheets after uniaxial and biaxial tensions.

Membrane Type	Biaxial Tension			Uniaxial Tension
	*E_M_* (GPa)	*σ_M_* (GPa)	*ε_M_* (%)	*v*
PG	839.613 ± 5.751	72.520 ± 0.471	9.075 ± 0.084	0.50 ± 0.01
PG-TP1	315.139 ± 4.267	31.369 ± 0.249	10.350 ± 0.097	0.29 ± 0.01
PG-TP2	366.552 ± 4.154	43.012 ± 0.283	11.575 ± 0.106	0.51 ± 0.01
PG-TP3	269.873 ± 3.967	33.368 ± 0.262	12.325 ± 0.114	0.69 ± 0.02

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
