# Peer review of "Theoretical Prediction of Mechanical Strength and Desalination Performance of One-Atom-Thick Hydrocarbon Polymer in Pressure-Driven Separation"

_polymers, 2019, doi:10.3390/polym11081358_

Round 1

Reviewer 1 Report

I very much enjoyed reading this manuscript. My expertise lies in molecular simulation and separation process, but I have not studied membranes outside of a classroom setting. Throughout the manuscript you do an excellent job motivating and explaining your work. It was very easy for me to follow along and appreciate exactly what you did. Excellent job! I offer just a few comments:

The work was all performed with available software (Dmol and LAMMPS are mentioned). I would therefore request that you make input files and force field files used by the software available with your supporting information documentation. The first time reading the "Computational details" section, I had made several notes with regards to computational details in the "Tensile Tests" subsection. Comments such as did you use commercial software? Can you provide input files? Can you provide additional force field details? You cited the ReaxFF paper, but is the implementation dependent on your software package. How did you treat long range interactions/long range corrections? Is a 2D lattice (Ewald type) method implemented to best handle inhomogeneous systems? Some of these details were better filed in in "Reverse Osmosis Desalination", but many remained (and it led to some more). In that subsection you did make note of Table S2, but then it is not until the very end of the manuscript that you mention additional computational details being available in the supporting information.

Now I understand that most readers will not be interested in these details. But some will be, especially those performing similar studies, and it is also very important that you work be reproducible. In my opinion a very simple solution would be to include input files and force field files, as used by the actual software, in the supporting information. You can just sort them in a folder, zip it, and include it as SI. Then just add a readme type doc to help the user understand what everything is. Then for the interested reader, there is absolutely not question what you did. And for the common reader not interested, much of this excess material stays out of the manuscript.

Page 8, line 241. You make a very nice, practical point that the kinetic diameter of water is approximately 2.8 A, which can be used to help interpret some of you findings. Do you have a reference for this value?

In the supporting information, table 2 needs to be updated. The phrase "Table 3" randomly appears in the middle of the table.

Author Response

Response to Reviewer 1

In this point-by-point response to the Reviewers’ comments:

Bold - response to the comments italic – revisions in the manuscript

Comments such as did you use commercial software? Can you provide input files? Can you provide additional force field details? You cited the ReaxFF paper, but is the implementation dependent on your software package. How did you treat long range interactions/long range corrections? Is a 2D lattice (Ewald type) method implemented to best handle inhomogeneous systems? Some of these details were better filed in "Reverse Osmosis Desalination", but many remained (and it led to some more). In that subsection you did make note of Table S2, but then it is not until the very end of the manuscript that you mention additional computational details being available in the supporting information.

Re: Thanks for your comments.

In this paper, the model construction and optimization were completed by Materials studio. Then the optimized model was converted into a readable data file of Lammps for simulation study. We added a note on lines 87, 106 and 117 of the manuscript.

Line 76-86 (Manuscript):

Classical molecular dynamics (MD) simulations were employed to perform biaxial (for the determination of fracture stress, elongation at break and Young’s modulus) and uniaxial (for the determination of Poisson’s ratio) tensile tests of PG-TPX membranes. The number of TP-like unit (defined as X) ranges from 0 to 3, leading to membrane structures with different pore sizes and periodically missing hexagons (Figure 1a-d). The membrane models with an initial dimension of approximately 100 ´ 100 Å2 in the x-y plane were located in the middle of the simulation box, while a vacuum slab was added along the z direction to make a box with the height of 80 Å. Periodic boundary conditions were applied in all three directions to eliminate the boundary effect of the atomic small-size membrane. Figure 1e and 1f show the tensile test along armchair and zigzag directions of the membrane, respectively. z direction remains unchanged during tension. All the above operations were carried out by Materials Studio.

Line 92-105 (manuscript):

To properly describe the bond association/disassociation events during the tensile process, reactive force field (ReaxFF)[35] was adopted. Moreover, to make sure our simulations are not biased by different potential versions, two different sets of parameters (i.e., Budzien et al. and Strachan et al.)[36, 37] were accepted. The mechanical parameters (i.e., tensile stress, fracture strain, Young’s module and Poisson’s ratio) from the two potentials were in excellent agreement (see Figure S1 in Supporting Information (SI) for the details). Before loading process of strain, structural optimizations of the initial PG-TPX materials were performed to release the in-plane residual stresses. Biaxial strain tests were performed in the NVT ensemble using a Nosé-Hoover thermostat[38] at 300 K with a damping factor of 100 time steps, whereas a Nosé-Hoover barostat at zero pressure was additionally applied on the non-enforced-strained boundary in the uniaxial strain tests (i.e., in the NPT ensemble) with a damping factor of 1000 time steps. A strain increment per 5000 steps at a rate of 5x10-7 fs-1 was applied for both types of tensile simulations. The time step was set to 0.1 fs. The stress at a given strain was obtained by averaging the stress in the last 2500 steps before the next strain increment. All continuum mechanics simulations were carried out using the open-source LAMMPS package[39].

Line 107-117 (Manuscript):

We constructed a combined model to assess the desalination performance of the membrane, i.e., water permeability and salt rejection. Figure 2 exhibits the model which consists of two rigid graphene pistons on the two ends, a membrane in the middle, and salt water and fresh water between the membrane and the two pistons. The membrane and the two pistons have been relaxed by Dmol3 using density functional theory (DFT) (lattice parameters can be found in Table S1). The box dimension in the xy plane is 45 × 35 Å2, and a large vacuum slab has been added to the z direction to provide enough mobile space for the piston wall. Periodic boundary conditions were applied in all three directions. 18 Na+/Cl- ion pairs were randomly distributed into 1800 water molecules to construct the salt water region, while the fresh water region was packed with 800 water molecules. Seawater desalination models were established in Materials Studio.

The input files and force fields related to the simulations have been uploaded as attachments. Welcome to download and view.

The ReaxFF is not dependent on our software package. Other softwares can also use this force field for calculations.

The long-range interactions were optimized by the particle-particle particle-mesh (PPPM) algorithm with a precision of 10-6e, which has been mentioned in line 125 in the manuscript.

We believe this method is efficient and effective, and it is suitable for two-dimensional system. The same method has been used in many previously reported simulation works similar to the current manuscript such as ACS Appl. Mater. Interfaces 2018, 10, 22, 18778−18786 and J. Phys. Chem. C 2014,118,13,6809-6819

In addition, in the revised paper, the mention of Table S2 has been moved to when the L-J parameters and atom charges are first stated.

Line 121-129 (Manuscript):

The nonbonding interactions between pairwise atoms were described by 6-12 Lennard-Jones (L-J) potentials and Coulombic interactions based on point charge models. The L-J potential parameters and charges for the systems are summarized in Table S2. The parameters for L-J potentials were taken from the DREIDING force field[39], and the pairwise interactions were truncated by a cutoff radius 12 Å. The atomic partial charges for Coulombic interactions were assigned by the charge equilibration (QEq) method[40], and the long-range interactions were optimized by the particle-particle particle-mesh (PPPM) algorithm with a precision of 10-6 e. Water molecules were modeled by the rigid SPC/E model[41] with the SHAKE[42] algorithm. The nonbonding potentials for salt ions and piston atoms were developed by Joung et al. [43] and Werder et al. [44], respectively.

Page 8, line 241. You make a very nice, practical point that the kinetic diameter of water is approximately 2.8 A, which can be used to help interpret some of your findings. Do you have a reference for this value?

Re: Thank you very much for the valuable advice. Reference (Crystal Research & Technology 2010, 21, 1299-1302) to the diameter of water molecular dynamics has been inserted into the manuscript.

Line 254-255 (Manuscript):

Theoretically, a water molecule has a kinetic diameter of approximately 2.8 Å[48], and therefore less likely to permeate through the membrane with a pore diameter smaller than or close to this value.

Line 434 (Manuscript):

Lilov, S. K. Determination of the Effective Kinetic Diameter of the Complex Molecules, Crystal Research & Technology 2010, 21, 1299-1302.

In the supporting information, table 2 needs to be updated. The phrase "Table 3. randomly appears in the middle of the table.

Re: Thanks for your comments. In the supporting information, the table S2 has been updated. As noted in the table, atomic charges assigned for the atoms in the piston or solution molecules can are provided. The phrase “Table S3” in the middle of the table indicates that atomic charges are provided separately in Table S3. We have annotated this in the titles of the two tables.

In addition, we added the reference sources of these L-J parameters and atomic charges to the Table S2.

Supporting Information:

Table S2. Non-bonding interaction parameters adopted in this work. Charges for atoms in the piston and solution can be found here. Charges for atoms in the studied PG structures can be found in Table S3, as noted in the table.

Element

ε (kcal/mol)

σ (Å)

q (e)

Ref.

Cpiston

0.0565

3.2140

0.0000

[4]

Cmembrane

0.0951

3.4730

Table S3

[5]

Hmembrane

0.0152

2.8464

[5]

Owater

0.1553

3.1660

-0.8476

[6]

Hwater

0.0000

0.0000

0.4238

[6]

Na+

0.3526

2.1595

1.0000

[7]

Cl-

0.0128

4.8305

-1.0000

[7]

Table S3. Atomic partial charges derived from DFT calculations for atoms in studied PG structures. The definition of atom types for each structure can be found in Figure S1.

Structure

qC1 (e)

qC2 (e)

qC3 (e)

qC4 (e)

qC5 (e)

qC6 (e)

qC7 (e)

qH1 (e)

qH2 (e)

qH3 (e)

qH4 (e)

(a) PG

-0.0009

-0.0606

0.0614

(b) PG-TP1

-0.0828

0.0187

0.0166

-0.0816

- 0.0902

0.0418

0.0707

0.0629

0.0646

(c) PG-TP2

0.0185

-0.0828

0.0171078

-0.0842

-0.0907

0.0421

-0.0908

0.0643

0.0638

0.0708

0.0714

(d) PG-TP3

0.0173

-0.0840

0.0422

-0.0909

0.0640

0.0713

Werder, T.; Walther, J. H.; Jaffe, R. L.; Halicioglu, T.; Koumoutsakos, P. On the Water−Carbon Interaction for Use in Molecular Dynamics Simulations of Graphite and Carbon Nanotubes, J. Phys. Chem. B, 112, 14090-14090. Mayo, S. L.; Olafson, B. D.; Goddard, W. A. DREIDING: a generic force field for molecular simulations, J. Phys. Chem. 1990, 94, 8897-8909. Berendsen, H. J. C.; Grigera, J. R.; Straatsma, T. P. The missing term in effective pair potentials, J. Phys. Chem. 1987, 91, 6269-6271. Joung, I. S.; Cheatham, Thomas E. Determination of Alkali and Halide Monovalent Ion Parameters for Use in Explicitly Solvated Biomolecular Simulations, J. Phys. Chem. B 2008, 112, 9020.

Reviewer 2 Report

The manuscript entitled “Mechanical Strength and Desalination Performance  of One-Atom-Thick Hydrocarbon Polymer as  Pressure-Driven Separation Membrane” Hu et al. is a theoretical work focused on atom thick membranes. The title of the manuscript is misleading, as the Authors neither mesure mechanical strenght nor found performance for any membranes. The reported results are theoretical predictions only and this should be clearly stated in the title.

 Similarly as in the case of the title, the Authors often refer to their results as facts rather than predictions. This is misleading and it should be underlined that these results are purely theoretical rather than experimental.

The main point is that there is no comparison of the calculation results with any experimental data. Without experimental data, the article is absolutely speculative. Perhaps it should be redirected to a more specialized journal, for example, MDPI Computation (section Computational chemistry covers Molecular dynamics,  Electronic structures, Density functional theory, Design and characterization of materials)

Author Response

Response to Reviewer 2

In this point-by-point response to the Reviewers’ comments:

Bold - response to the comments italic – revisions in the manuscript

1- The manuscript entitled “Mechanical Strength and Desalination Performance of One-Atom-Thick Hydrocarbon Polymer as Pressure-Driven Separation Membrane” Hu et al. is a theoretical work focused on atom thick membranes. The title of the manuscript is misleading, as the Authors neither measure mechanical strength nor found performance for any membranes. The reported results are theoretical predictions only and this should be clearly stated in the title.

Re: Thanks for this valuable suggestion. We have revised the title of the article as “Theoretical Prediction of Mechanical Strength and Desalination Performance of One-Atom-Thick Hydrocarbon Polymer in Pressure-Driven Separation”. We think the current title clearly states the employment of theoretical method throughout the whole work.

2- Similarly as in the case of the title, the Authors often refer to their results as facts rather than predictions. This is misleading and it should be underlined that these results are purely theoretical rather than experimental.

Re: Thanks for this comment. We have revised the conclusions according to your suggestion. In the conclusion, we have pointed out that the results are obtained by the simulations and have certain predictability.

Line 166-167 (Manuscript):

So we predicted this makes PG-TP2 hard to be deformed and thus exhibits stronger mechanical strength.

Line 223-224 (Manuscript):

In other words, the theoretical predictions indicate that 2D hydrocarbon polymers PG-TP2 designed in this work supported by the substrate materials with the appropriate pore size can withstand hydraulic pressure for practical separation application without brittle fracture.

Line 289-292 (Manuscript):

Therefore, the simulation results indicate that it could potentially act as a high-permeability RO membrane for one-step water desalination. Moreover, owing to the relatively lower salt rejection, PG-TP3 could provide opportunities in brackish water desalination or multi-stage seawater desalination.

Line 301-303 (Manuscript):

Moreover, the simulation results predict that one-atom-thick hydrocarbon polymers have better water desalination performance compared with the traditional polymeric reverse osmosis membranes.

The main point is that there is no comparison of the calculation results with any experimental data. Without experimental data, the article is absolutely speculative. Perhaps it should be redirected to a more specialized journal, for example, MDPI Computation (section Computational chemistry covers Molecular dynamics, Electronic structures, Density functional theory, Design and characterization of materials)

Re: We thank the reviewer for the important suggestion. However, in the current work we focus on the study of a one-atom-thick hydrocarbon polymer material. We think this paper is suitable for Polymers as the journal aims to publish papers which advance the fields of (i) polymerization methods, (ii) theory, simulation, and modeling, (iii) understanding of new physical phenomena, (iv) advances in characterization techniques, and (v) harnessing of self-assembly and biological strategies for producing complex multifunctional structures. Our work will be instrumental to the design of new membranes and the evaluation of their properties for pressure-driven separation applications, which meets the (ii) and (iv) requirements well. Even though we use simulation methods, it would not make big influence for the publication as long as our work meets the requirement of the journal. Again thank you very much for your comment.

Reviewer 3 Report

The manuscript is presenting an interesting approach for the membrane dedicated to desalination process by pressure-driven process. The manuscript is organized and prepared correctly. However, editing correction is needed.

Considering the topic of the work is slightly similar to already published research. Most important in the presented manuscript is unfortunately quite a lot of text copied from different works, e.g. Q. Lyu et al. Rational Design of Two-Dimensional Hydrocarbon Polymer as Ultrathin-Film Nanoporous Membranes for Water Desalination, ACS Appl. Mater. Interfaces2018102218778-18786; D. Cohen-Tanugi, J.C. Grossman, Mechanical Strength of Nanoporous Graphene as a Desalination Membrane Nano Lett.201414116171-6178. The authors need to highlight the novelty of the work referring to the already published work.

What type of real problem the presented approach can solve?

There is a real necessity to prepare the atomic level membranes? It is possible to scale-up the membrane?

Considering the presented data, if possible please add the standard deviation to table 1. The quality of the figures is good and appropriate for the journal.

The presented literature is slightly out-of-date, only 8% of the references are from the last 5 years. Better literature survey should be done to check that something newer is available.   

Author Response

Response to Reviewer 3

In this point-by-point response to the Reviewers’ comments:

Bold - response to the comments italic – revisions in the manuscript

Considering the topic of the work is slightly similar to already published research. Most important in the presented manuscript is unfortunately quite a lot of text copied from different works, e.g. Q. Lyu et al. Rational Design of Two-Dimensional Hydrocarbon Polymer as Ultrathin-Film Nanoporous Membranes for Water Desalination, ACS Appl. Mater. Interfaces2018102218778-18786; D. Cohen-Tanugi, J.C. Grossman, Mechanical Strength of Nanoporous Graphene as a Desalination Membrane Nano Lett.201414116171-6178. The authors need to highlight the novelty of the work referring to the already published work.

Re: Thanks for your comment. The study of Lyu et al mainly illustrates the design of pore structure in a monatomic layer membrane material which can be used for seawater desalination. The study of D. Cohen-Tanugi mainly demonstrates the influence of porosity and substrate on the mechanical properties of porous graphene. Here, we investigated the mechanical strength of two-dimensional hydrocarbon polymers containing sub-nanometer pores with various topologies. We demonstrated that the mechanical strength of the membranes is correlated with their pore sizes and geometries. In addition, when the pore size of the substrate is controlled within a reasonable range, all the studied PG membranes can withstand the practical hydraulic pressure of few megapascals. Results in relation to above discussions have not been previously reported. Therefore, we think our work is quite different from these literatures. To highlight the novelty of the work referring to the already published work, we make a further discussion on the findings in the work in lines 236-241.

Line 234-239 (Manuscript):

In this prat, we investigated the mechanical strength of two-dimensional hydrocarbon polymers containing sub-nanometer pores with various topologies. We demonstrated that the mechanical strength of the membranes is correlated with their pore sizes and geometries. In addition, when the pore size of the substrate is controlled within a reasonable range, all the studied PG membranes can withstand the practical hydraulic pressure of few megapascals. Results in relation to above discussions have not been previously reported.

What type of real problem the presented approach can solve?

Re: The studied materials exhibit better seawater desalination performance as compared to the traditional polymeric reverse osmosis membrane. This work could be instrumental to designing new membranes and evaluating their properties for pressure-driven separation applications.

There is a real necessity to prepare the atomic level membranes? It is possible to scale-up the membrane?

Re: Thanks for your considered comment on this point. It is currently unable to prepare the atomic level membranes. However, we think it will be a breakthrough it this can be achieved considering the astonishing properties of membranes at the atomic level. The molecular dynamics simulation and continuum mechanics simulation adopted in this paper focus on the microscopic level properties of membranes. The simulation results of the single atomic layer model can predict the influence law in practical application, which has certain guiding significance for the experiment. Due to the computational capability, the constructed model may be limited. We think the membranes can be scaled up considering the real layer-by-layer accumulation of membranes for practical needs. This may be another important point that can be fully explored by researchers in the future. Our work mainly illustrates the mechanisms at atomic-level that are difficult to be revealed by pure experiments. Many previously reported works have been carried out using similar method. we provide four published papers talking about this issue.

Lyu et al. ACS Appl. Mater. Interfaces 2018, 10,22, 18778-18786; Cohen-Tanugi et al. Nano Lett. 2014, 1411, 6171-6178;

K Zhang et al. Environ. Sci.: Water Res. Technol. 2017, 10,1039;

Li-Chiang Lin et al. Chem. Comm. 2015, 00, 1-3

4.- Considering the presented data, if possible please add the standard deviation to table 1. The quality of the figures is good and appropriate for the journal.

Re: Thank you very much for your comment. The standard deviations of the data in Table 1 have been added.

Line 171(Manuscript):

Table 1. Mechanical parameters of PG-TPX sheets after uniaxial and biaxial tensions.

membrane type

biaxial tension

uniaxial tension

EM (GPa)

σM (GPa)

εM (%)

v

PG

839.613±5.751

72.520±0.471

9.075±0.084

0.50±0.01

PG-TP1

315.139±4.267

31.369±0.249

10.350±0.097

0.29±0.01

PG-TP2

366.552±4.154

43.012±0.283

11.575±0.106

0.51±0.01

PG-TP3

269.873±3.967

33.368±0.262

12.325±0.114

0.69±0.02

5.- The presented literature is slightly out-of-date, only 8% of the references are from the last 5 years. Better literature survey should be done to check that something newer is available. 

Re: Thanks for your comment. Through literature survey, we have updated some references in lines 30-32. The list of newly added references is as follows:

Line 28-37(Manuscript):

Due to the characteristics of high efficiency, energy conservation and environmental friendliness, membrane-base separation technology has been applied to the purification of biofuel extraction, water desalination and sewage treatment[1-6]. To data, graphene oxides[7], covalent organic frameworks(COFs)[8, 9], metal organic frameworks(MOFs)[10-12], zeolite[13], carbon nano tube (CNT)[14, 15] and other advanced materials are considered as separation membrane materials. Among numerous membrane materials, nanoporous ultrathin-film materials such as one-atom-thick porous graphene (PG) have drawn considerable attention in the scientific community for their potential as the next generation of separation membrane materials[16-23]. To date, PG has been prepared by drilling nanopores in original graphene sheets, denoted as the so-called top-down approach.[24-27]

Line 334-357(Manuscript):

Yuan, L.; Miao, T.; Rong, W. A high-performance and robust membrane with switchable super-wettability for oil/water separation under ultralow pressure, J. Member. Sci. 2017, 543, 123-132. Fathizadeh, M.; Xu, W. L.; Zhou, F.; Yoon, Y.; Miao, Y. Graphene Oxide: A Novel 2‐Dimensional Material in Membrane Separation for Water Purification, Adv. Mater. Interface 2017, 4, 1600918. Yuan, S.; Li, X.; Zhu, J.; Zhang, G.; Bruggen, B. V. D. Covalent organic frameworks for membrane separation, Chem. Soc. Rev. 2019, 48, 2665-2681. Lin, L. C.; Choi, J.; Grossman, J. C. Two-dimensional covalent triazine framework as an ultrathin-film nanoporous membrane for desalination, Chem. Commun. 2015, 51, 14921-14924. Kadhom, M.; Deng, B. Metal-organic frameworks (MOFs) in water filtration membranes for desalination and other applications, Appl. Mat. Today 2018, 11, 219-230. Sun, H.; Tang, B.; Wu, P. Development of hybrid ultrafiltration membranes with improved water separation properties using modified super-hydrophilic metal-organic framework nanoparticles, ACS Appl. Mat. Interfaces 2017, 9, 21473. Kadhom, M.; Hu, W.; Deng, B. Thin Film Nanocomposite Membrane Filled with Metal-Organic Frameworks UiO-66 and MIL-125 Nanoparticles for Water Desalination, Membranes 2017, 7, 31. Yong, P.; Lu, H.; Wang, Z.; Yan, Y. Microstructural optimization of MFI-type zeolite membranes for ethanol–water separation, J. Mater. Chem. A 2014, 2, 16093-16100. Hsieh, C. T.; Hsu, J. P.; Hsu, H. H.; Lin, W. H.; Juang, R. S. Hierarchical oil–water separation membrane using carbon fabrics decorated with carbon nanotubes, Surf. Coat. Technol. 2016, 286, 148-154. Khalid, A.; Al-Juhani, A. A.; Al-Hamouz, O. C.; Laoui, T.; Khan, Z.; Atieh, M. A. Preparation and properties of nanocomposite polysulfone/multi-walled carbon nanotubes membranes for desalination, Desalination 2015, 367, 134-144.

Round 2

Reviewer 3 Report

The manuscript has been corrected. The authors took into account all comments and suggestion. Just please double-check the spelling, for instance, line 234 "in this prat" should be "part". Please, correct.